# Overexpression of ATase1 and ATase2 disrupts the secretome and causes a progeria phenotype

Tzu-Lin Cheng[1,2,3,]*, Feixuan Wu[4,]* , Md Ezazul Haque[1,2], Abigail R Thiel[2], Danqing Wang[4] , Jeffrey J Helgager[5], Lingjun Li[4,6] , Luigi Puglielli[1,2,7,8]

Nε-lysine acetylation in the lumen of the ER requires two acetyltransferases, ATase1/NAT8B and ATase2/NAT8. They are type II membrane proteins and belong to the larger GNAT superfamily of acetyltransferases. Their enzymatic activity is tightly coupled to the import of acetyl-CoA in the lumen of the ER by AT-1/SLC33A1. Gene duplication events involving 3q25.31 (harboring AT-1/SLC33A1) and 2p13.1 (harboring ATase1/NAT8B and ATase2/NAT8) are associated with autism spectrum disorder with intellectual disability and progeria-like dysmorphism. Here, we report the generation and phenotypic characterization of mice with systemic overexpression of ATase1 (ATase1 sTg) and ATase2 (ATase2 sTg). Overexpression of either ATase at conception was found to be lethal while overexpression at birth was found to cause a progeria-like phenotype that included skin alterations, lordokyphosis, reduced bone density, sarcopenia, splenomegaly, adenomegaly, and systemic inflammation. The phenotype of ATase1 sTg mice displayed incomplete penetrance, while the phenotype of ATase2 sTg displayed full penetrance and was more severe. Mechanistically, the phenotype was linked to altered dynamics of the secretory pathway with defects affecting the quality of the secretome.

## Introduction

Nε-lysine acetylation is a highly dynamic post-translational event that regulates a variety of protein functions (Puglielli et al, 2023). In the lumen of the ER, Nε-lysine acetylation is carried out by ATase1/NAT8B and ATase2/NAT8 (Fernandez-Fuente et al, 2023b). Both ATases are type II membrane proteins; the catalytic domain is exposed to the lumen of the organelle and displays the conserved R/Q-x-x-G-x-G/A acetyl-CoA binding motif of the GNAT superfamily

of Nε-lysine acetyltransferases (Fernandez-Fuente et al, 2023b). The activity of the ATases is tightly coupled to the influx of acetyl-CoA from the cytosol, which is ensured by AT-1/SLC33A1, an ER membrane transporter that acts as an antiporter by coupling the cytosol-to-ER transfer of acetyl-CoA with the ER-to-cytosol exit of free CoA (Fernandez-Fuente et al, 2023b).

Defective ER acetylation, as caused by either mutations or gene duplication events, is associated with severe rare human diseases (Fernandez-Fuente et al, 2023b). Importantly, duplications involving 3q25.31 (harboring AT-1/SLC33A1) and 2p13.1 (harboring ATase1/NAT8B and ATase2/NAT8) are specifically associated with autism spectrum disorder with intellectual disability and progeria-like dysmorphism (see National Organization for Rare Disorders database; see also [Francke, 1978; Fineman et al, 1983; Fryns et al, 1989; Sawyer et al, 1994; Rizzu et al, 1997; Ounap et al, 2005; Krumm et al, 2013; Poultney et al, 2013; Krumm et al, 2015]). In the case of AT-1, the disease association was successfully modeled (and validated) in the mouse. Indeed, mice knock-in for a loss-of-function AT-1 mutation associated with spastic paraplegia developed a peripheral form of neuropathy (Peng et al, 2014; Liu et al, 2017), mice with neuron-specific overexpression of AT-1 developed an autistic-like phenotype (Hullinger et al, 2016), and mice with systemic overexpression of AT-1 developed a progeria-like phenotype with reduced lifespan (Peng et al, 2018). However, the disease association with the ATases is yet to be validated in a model organism.

Substrates of ATase1 and ATase2 include both ER-resident and -trafficking proteins (Pehar et al, 2012; Rigby et al, 2021). ER-resident proteins that are modified by the ATases are involved in a variety of functions, including regulation of folding and quality control within the ER lumen, selection of correctly folded proteins to be transported to the Golgi apparatus, and activation of ER-specific autophagy (Farrugia & Puglielli, 2018; Fernandez-Fuente et al, 2023b). In essence, the ATases are predicted to influence functional dynamics of the secretory pathway by regulating protein

[1]Department of Medicine, University of Wisconsin-Madison, Madison, WI, USA   [2]Waisman Center, University of Wisconsin-Madison, Madison, WI, USA   [3]Neuroscience Training Program, University of Wisconsin-Madison, Madison, WI, USA   [4]School of Pharmacy, University of Wisconsin-Madison, Madison, WI, USA   [5]Department of Pathology and Laboratory Medicine, University of Wisconsin-Madison, Madison, WI, USA   [6]Department of Chemistry, University of Wisconsin-Madison, Madison, WI, USA   [7]Geriatric Research Education Clinical Center, Veterans Affairs Medical Center, Madison, WI, USA   [8]Department of Neuroscience, University of Wisconsin-Madison, Madison, WI, USA

Correspondence: lingjun.li@wisc.edu; lp1@medicine.wisc.edu
*Tzu-Lin Cheng and Feixuan Wu contributed equally to this work

homeostasis (proteostasis) within the ER/secretory pathway. The implication of the ATases in the regulation of ER-specific authophagy was successfully tested in Atase1$^{-/-}$ and Atase2$^{-/-}$ mice (Rigby et al, 2021). However, the physiological consequences of ATase overexpression are yet to be tested.

Here, we report the generation and phenotypic characterization of mice with systemic overexpression of ATase1 (ATase1 sTg) or ATase2 (ATase2 sTg). Overexpression of either ATase at conception was found to be lethal while overexpression at birth was found to cause a progeria-like phenotype that included skin alterations, lordokyphosis, reduced bone density, sarcopenia, splenomegaly, adenomegaly, and systemic inflammation. Interestingly, the phenotype of ATase1 sTg mice displayed incomplete penetrance and was less severe, while the phenotype of ATase2 sTg displayed full penetrance and was more severe.

Although they share enzymatic activity and can compensate each other in vivo, the ATases are regulated differently, and differently influence the activation of autophagy as well as the global acetyl-CoA metabolic response (Rigby et al, 2020, 2021). Here, we took advantage of these new mouse models and were able to dissect their specific biological roles. Importantly, overexpression of the ATases altered the ER-to-Golgi transition of nascent glycoproteins and forced selective and divergent adaptive responses outside of the ER.

In conclusion, this study highlights specific and fundamental biological functions of ER acetylation and reveals novel adaptive responses to ER-specific N$\varepsilon$-lysine acetylation within the cell that impact the pathophysiology of associated human diseases.

# Results

### ATase1 sTg and ATase2 sTg mice display a progeria-like phenotype

To study the systemic effect of ATase1 and ATase2 overexpression, we used an inducible Tet-Off system driven by the Rosa26 locus that can be inactivated with doxycycline (Peng et al, 2018; Fernandez-Fuente et al, 2023a). Overexpression of ATase1 from conception (ATase1 sTg$^{OC}$) yielded less than 10% of double transgenic mice, which is below the expected Mendelian ratio. Of them, about 20% developed a progeria-like phenotype while the remaining 80% displayed no significant disease manifestations throughout our study (Fig 1A and B). Administration of doxycycline to the pregnant females restored the Mendelian ratio, thus confirming the embryonic lethality of ATase1 systemic overexpression (Fig 1C). When ATase1 was overexpressed at birth (ATase1 sTg$^{OB}$), by removing doxycycline from the diet, about 75% of the double transgenic animals developed a progeria-like phenotype with delayed growth that was indistinguishable from the phenotype of ATase1 sTg$^{OC}$ mice (Fig 1B–D). ATase1 sTg$^{OB}$ animals (simply referred to as ATase1 sTg thereafter) were used as the main model for our study. Evidence of systemic overexpression of human ATase1 was documented by Western blot (Fig 1E). At humane endpoint, which occurred around the age of 1 yr, ATase1 sTg mice displayed reduced body weight (Fig 1D), hair loss with skin lesions (Fig 1C),

cardiomegaly (Fig 2A), splenomegaly (Fig 2A and B), adenomegaly (Fig 2B), systemic inflammation (Fig 2C), reduced bone density (Fig 2D), rectal prolapse, and modest sarcopenia (Table 1). When we analyzed the levels of the senescent markers p16, p21, and $\beta$-Gal, we only observed a modest up-regulation of $\beta$-Gal staining (Fig 2E and F).

Next, we overexpressed human ATase2 using the same Rosa26 system (Fig 3A). As with ATase1 sTg$^{OC}$, systemic overexpression of ATase2 from conception (ATase2 sTg$^{OC}$) was embryonically lethal (Fig 3B). Two main differences were observed between ATase1 sTg$^{OC}$ and ATase2 sTg$^{OC}$. First, the effect of ATase2 overexpression was fully penetrant; no live double transgenic mouse was ever obtained. Second, we recovered incompletely developed dead fetuses from ATase2 sTg$^{OC}$ females but not from ATase1 sTg$^{OC}$ females. This could suggest that the two ATases have partially distinct developmental roles with the overexpression of ATase1 affecting early developmental events, albeit with incomplete penetrance, and overexpression of ATase2 affecting late developmental events.

When ATase2 was overexpressed at birth (ATase2 sTg$^{OB}$), by removing doxycycline from the diet, all the double transgenic animals developed a progeria-like phenotype (Fig 3C). ATase2 sTg$^{OB}$ animals (simply referred to as ATase2 sTg thereafter) were used as the main model for our study. Within a few weeks, ATase2 sTg mice developed hair loss and skin lesions, and by the age of ~2 mo they all manifested a severe progeria-like phenotype (Fig 3C). The animals also displayed growth delay and a short lifespan (Fig 3C–E). Evidence of systemic overexpression of human ATase2 was documented by Western blot (Fig 3F). The phenotype of ATase2 sTg mice included histological alterations of the skin (Fig 4A), rectal prolapse (Fig 4B), splenomegaly (Fig 4C), adenomegaly (Fig 4C), hepatomegaly (Fig 4D), reduced bone density (Fig 4E), sarcopenia, systemic inflammation (Fig 4F), and upregulation of $\beta$-Gal staining (Fig 4G), which was observed in the absence of increased p16 and p21 markers (Fig 4H; see also Table 1). Finally, about 40% of ATase2 sTg mice developed tumor-like lesions histologically compatible with epidermal inclusion cysts in the area corresponding to the parotid and submandibular gland region (Fig 4I).

In conclusion, systemic overexpression of either ATase resulted in a phenotype that was reminiscent of segmental forms of progerias (Table 1) (Pivnick et al, 2000; Liao & Kennedy, 2014; Gonzalo et al, 2017; Karikkineth et al, 2017). When compared with ATase1 sTg mice, the phenotype of ATase2 sTg mice was fully penetrant and always more severe.

### ATase1 sTg and ATase2 sTg mice display widespread proteomic changes

To elucidate the impact of increased ATase activity on protein expression, we conducted large-scale liquid chromatography coupled to tandem mass spectrometry (LC-MS/MS) analysis to identify global changes in the proteome of ATase1 sTg and ATase2 sTg animals. Since the ER acetylation machinery is heavily involved in dynamics of the secretory pathway (Fernandez-Fuente et al, 2023b), we performed our study with brain cortical and hippocampal tissue. In the cortex, we found 156 and 139 proteins

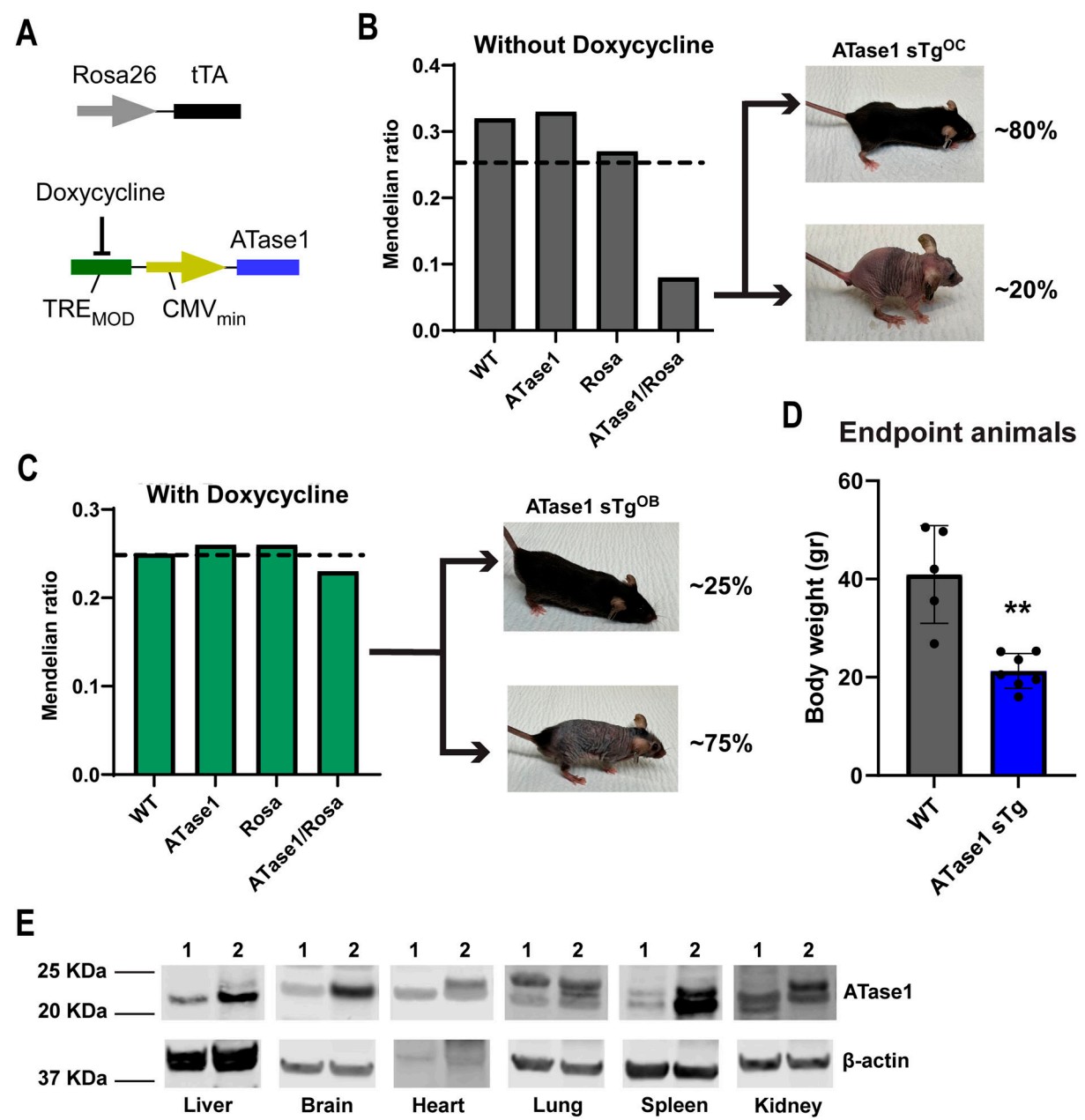

**Figure 1. Systemic overexpression of human ATase1 caused a progeria-like phenotype with incomplete penetrance.**
**(A)** ATase1 sTg mice were generated with a Tet-Off expression system driven by the Rosa26 locus. **(B)** Overexpression of ATase1 from conception reduced the yield of ATase1 sTg mice. The progeria-like phenotype of ATase1 sTg$^{OC}$ mice displayed incomplete penetrance. **(C)** Overexpression of ATase1 at birth restored the Mendelian ratio. The progeria-like phenotype of ATase1 sTg$^{OB}$ mice displayed incomplete penetrance. **(D)** ATase1 sTg mice with progeria-like phenotype displayed reduced body weight. The number of animals is shown. $**P < 0.005$. Welch't test. **(E)** Western blot showing overexpression of ATase1 in different organs (1. WT mice, 2. ATase1 sTg mice). Mice were ~1 yr old when studied.
Source data are available for this figure.

being altered in ATase1 and ATase2 sTg females, respectively (Fig 5A). The profile was quite different among the two models, with only 24 proteins being represented in both (Figs 5A and S1). A similar outcome was observed when we compared the males. Indeed, 187 and 223 proteins, respectively, were found altered with only 49 being represented in both models (Figs 5B and S2). In essence, overexpression of either ATase yielded widespread but significantly different proteomic changes in the brain of the two models (Fig S3). The list of significantly changed proteins in the cortex across models with relevant heatmaps is provided in Figs S1 and S2. STRING analysis at >90% confidence revealed four clusters in ATase1 sTg and six clusters in ATase2 sTg that correlated highly with specific cellular process (Fig 5C). They were translation/ribosomes, nucleosome core/histones, tRNA aminoacylation and

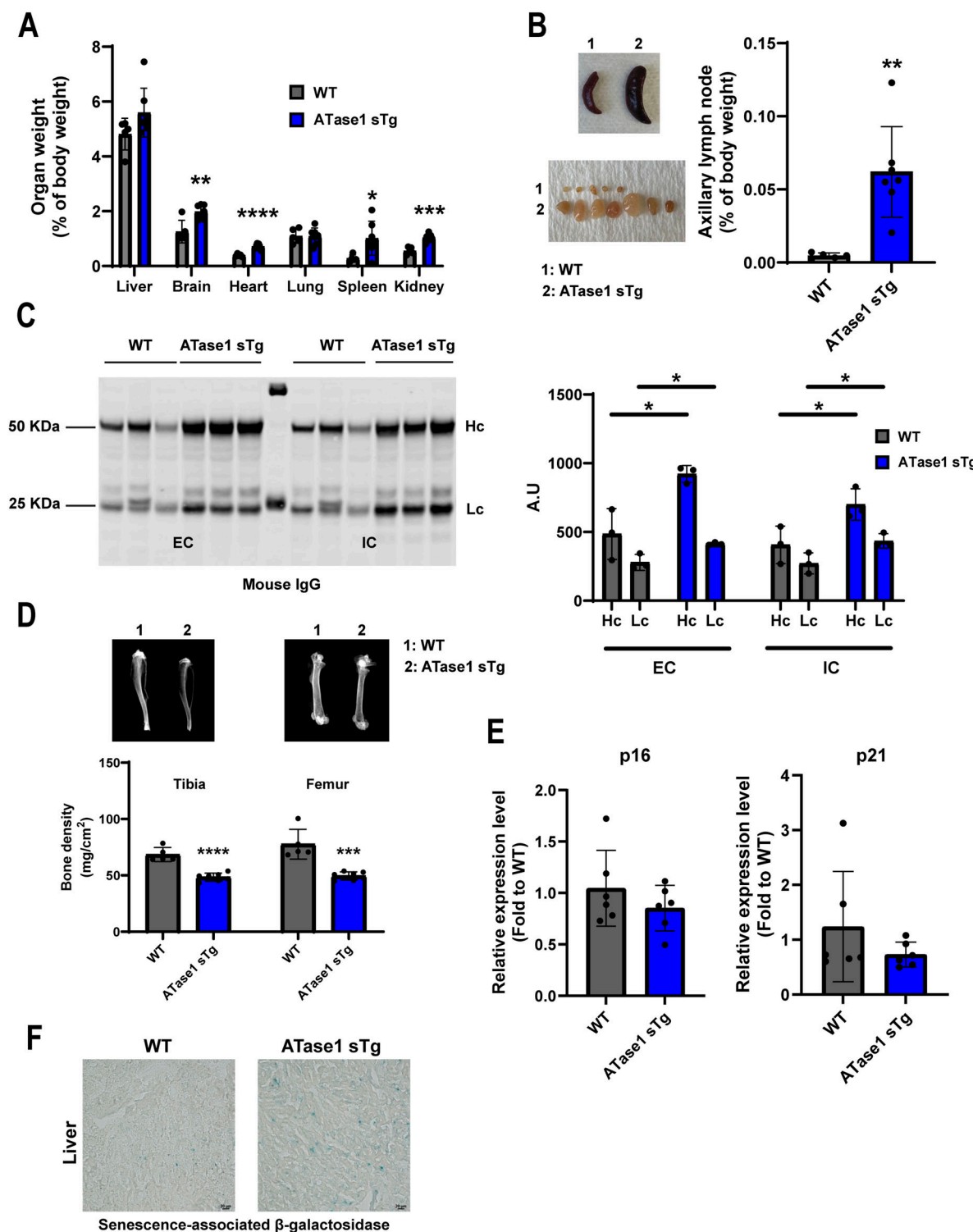

**Figure 2.  ATase1 sTg mice displayed progeria-like features.**
**(A)** Organ to body weight ratio. The number of animals is shown. *$P < 0.05$; **$P < 0.005$; ***$P < 0.0005$; ****$P < 0.0001$. Unpaired $t$ test or Welch't test. **(B)** ATase1 sTg mice displayed splenomegaly and adenomegaly (axillary lymph nodes are shown). The number of animals is shown. **$P < 0.005$. Welch't test. **(C)** ATase1 sTg mice displayed increased Ig tissue infiltration (liver). The left panel shows the Western blot while the right panel shows the associated quantification. The number of animals is shown. *$P < 0.05$. Unpaired $t$ test. EC, extracellular; IC, intracellular; Hc, heavy chain; Lc, light chain. **(D)** ATase1 sTg mice displayed reduced bone density (Tibia and Femur are shown). The number of animals is shown. ***$P < 0.0005$; ****$P < 0.0001$. Unpaired $t$ test or Welch't test. **(E)** Senescence markers p16 and p21 (brain). **(F)** Senescence associated $\beta$-galactosidase staining (liver). Mice were ~1 yr old when studied.
Source data are available for this figure.

**Table 1. Observed phenotype of ATase1 and ATase2 sTg mice**

| | ATase1 | ATase2 |
|---|---|---|
| Lifespan | Reduced | Reduced |
| Endpoint | 1 yr | 2–3 mo |
| Body weight | Reduced | Reduced |
| Hunched posture | Observed | Observed |
| Hair loss | Observed | Observed |
| Skin lesion | Observed | Observed |
| Muscle atrophy | Observed | Observed |
| Adipose tissue | Reduced | Reduced |
| Bone density | Reduced | Reduced |
| Liver size | Normal | Increased |
| Heart size | Increased | Normal |
| Spleen size | Increased | Increased |
| Lymph node size | Increased | Increased |
| Rectal prolapse | Observed | Observed |
| Systemic inflammation | Observed | Observed |

ATase1 sTg only includes animals with phenotype.

acute-phase inhibitors/protease inhibitors in ATase1 sTg mice; and protein folding/quality control, ubiquitin/proteasome/ER stress, mitochondrial translation, fatty acid metabolism, cell adhesion/protease inhibitors, and tRNA aminoacylation in ATase2 sTg mice (Fig 5C). Clusters 1 (translation and ribosomes) and 3 (tRNA aminoacylation) in ATase1 sTg, and clusters 1 (protein folding and quality control) and 6 (tRNA aminoacylation) in ATase2 sTg mice are highly relevant to protein biosynthesis and regulation of the secretory pathway, a function that has been connected to the ER acetylation machinery. Cluster 2 (ubiquitin/proteasome/ER stress) in ATase2 sTg included proteins that are relevant to the ER-associated degradation system. Finally, both models displayed general metabolic outcomes, as highlighted by cluster 2 (nucleosome core/histones) in ATase1 sTg and clusters 3 (mitochondrial translation) and 4 (fatty acid metabolism) in ATase2 sTg mice (Fig 5C).

## ATase1 sTg and ATase2 sTg mice display widespread changes in the N-glycosylation profile of transiting proteins

The largest proteomic changes in both sTg models were found to be relevant for protein biosynthesis and regulation of the secretory pathway subgroups (Fig 6A–C). They included translation initiation factors, ribosomal associated proteins, enzymes necessary for folding, chaperones and quality control-associated proteins, as well as integral components of the oligosaccharyl transferase (OST) complex, which is responsible for the initial N-glycosylation of nascent glycoproteins within the ER lumen. Taken together, these data suggest that the ability of the ER to control the engagement of the secretory pathway and the quality of the *secretome* might be affected in the sTg animals.

Within the conventional secretory pathway, newly synthesized proteins that survive quality control in the ER translocate to the Golgi apparatus to complete post-translational maturation and then be delivered to their final destination. Importantly, N-glycosylation begins in the lumen of the ER where the OST transfers a pre-formed $GlcNAc_2Man_9Glc_3$ oligosaccharide structure onto an Asn residue within the N-X-S/T consensus motif (Fig 6D). This initial oligosaccharide undergoes major modifications as glycoproteins move out of the ER and through the Golgi apparatus. Specifically, the three terminal glucose are removed and the high-mannose structure is trimmed to allow final modification, which includes addition of fucose and galactose in the *cis/medial*-Golgi and sialic acid in the *trans*-Golgi and *trans*-Golgi network (Fig 6D) (Hirschberg et al, 1998). In essence, the structure of the oligosaccharide complex reflects the ability of the ER to deliver correctly folded N-glycoproteins to the Golgi apparatus as well as the ability of the Golgi apparatus to modify transiting N-glycoproteins.

Importantly, we previously reported that the ATases interact with the OST within a high-molecular mass complex suggesting an intimate connection between OST-dependent N-glycosylation and ATase-dependent N$\varepsilon$-lysine acetylation of newly synthesized N-glycoproteins (Ding et al, 2014). This argument is reinforced by the dataset included here (Fig 6C), as reflected by changes in the levels of Stt3b, Rpn1, and Ddost, which are integral components of the mouse Ost super-complex, as well as Alg9 and Alg10b, which are involved in the assembly of the initial $GlcNAc_2Man_9Glc_3$ oligosaccharide structure. Therefore, to determine whether the changes reflected by the proteome data impinge on the quality of the *secretome* by influencing the ER-to-Golgi transition of nascent proteins, we used sequential hydrophilic interaction chromatography (HILIC) to enrich intact N-glycopeptides and LC-MS/MS to analyze the composition of the oligosaccharides and the incorporation of Golgi-specific sugars. To be consistent with the design of the proteome, we characterized the glycoproteome of the brain.

We found significant changes in 145 and 98 glycoforms of ATase1 sTg and ATase2 sTg females, and 108 and 89 glycoforms of the ATase1 sTg and ATase2 sTg males (Fig 7A). As with the proteome, the overlap among models were minimal, with 22 glycoproteins/glycoforms shared between ATase1 sTg and ATase2 sTg females and 19 between ATase1 sTg and ATase2 sTg males (Fig S4). Most of the observed changes were accounted for by sialic acid, fucose, and high mannose and were clustered within plasma membrane, cell surface, synapse, extracellular, and vesicle subgroups (Fig 7B). In essence, Golgi-dependent events and cellular compartments that depend on successful transition across the Golgi apparatus and toward the plasma membrane were affected.

When we quantified all significant changes per glycosite, we found that most of the glycosites had one glycan structure affected with significant heterogeneity observed across models. Indeed ~20% of glycosites had more than one glycan structure being significantly affected (Fig 7C). Furthermore, when we examined the glycan-type distribution as they pertain to the number of sites affected per protein, we observed a higher appearance of fucose on multiple sites per protein with an almost even distribution of sialic acid (Fig 7D). Collectively, these findings highlight the complexity of the N-glycan modifications caused by the overexpression of the ATases and point to specific changes of individual glycosylation events occurring at the ER-to-Golgi transition as well as within the Golgi apparatus. Finally, analysis of all significantly

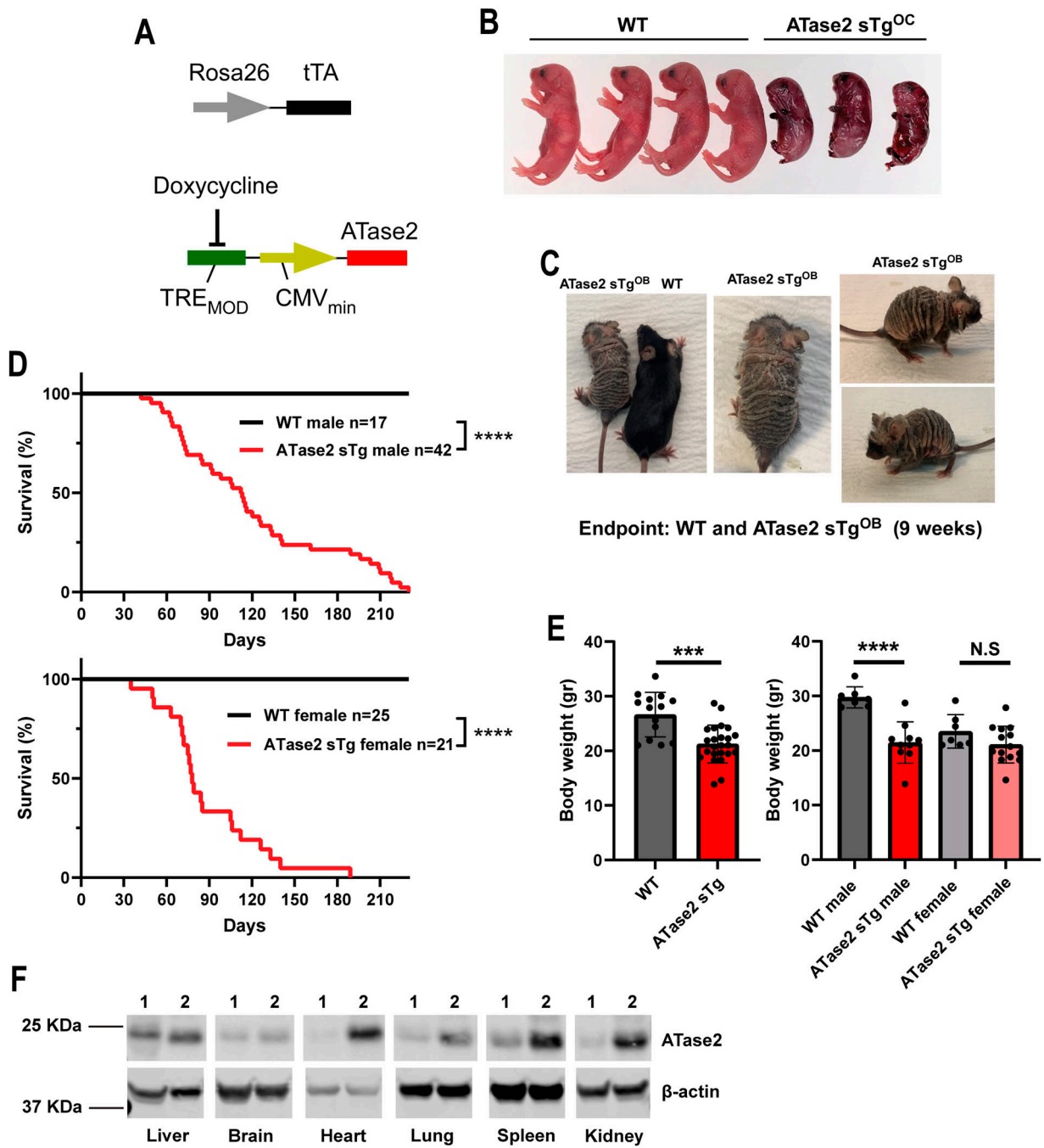

**Figure 3. Systemic overexpression of human ATase2 caused a progeria-like phenotype.**
**(A)** ATase2 sTg mice were generated with a Tet-Off expression system driven by the Rosa26 locus. **(B)** Overexpression of ATase2 from conception was embryonically lethal. Incompletely developed ATase2 sTg^OC are shown. **(C)** Overexpression of ATase2 at birth caused a fully penetrant progeria-like phenotype. One representative ATase2 sTg^OB and WT littermate are shown next to each other. **(D)** ATase2 sTg mice displayed short life span. Maximum lifespan: males = 230 d, females = 189 d. The number of animals is shown. ****$P < 0.0001$ via the Kaplan-Meier lifespan test. **(E)** ATase2 sTg mice displayed reduced body weight at humane endpoint. The number of animals is shown. ***$P < 0.0005$; ****$P < 0.0001$. Unpaired $t$ test. **(F)** Western blot showing overexpression of ATase2 in different organs (1. WT mice, 2. ATase2 sTg mice). Mice were ~3 mo old when studied.
Source data are available for this figure.

affected glycoproteins across the brain (cortex and hippocampus) points to fundamental biological functions that are intimately connected to neuronal biology and are highly dependent on dynamics of the secretory pathway (Fig S5 and S6). They included cell surface-dependent events, cell-cell adhesion/interactions, extracellular matrix organization, assembly and activity of synaptic

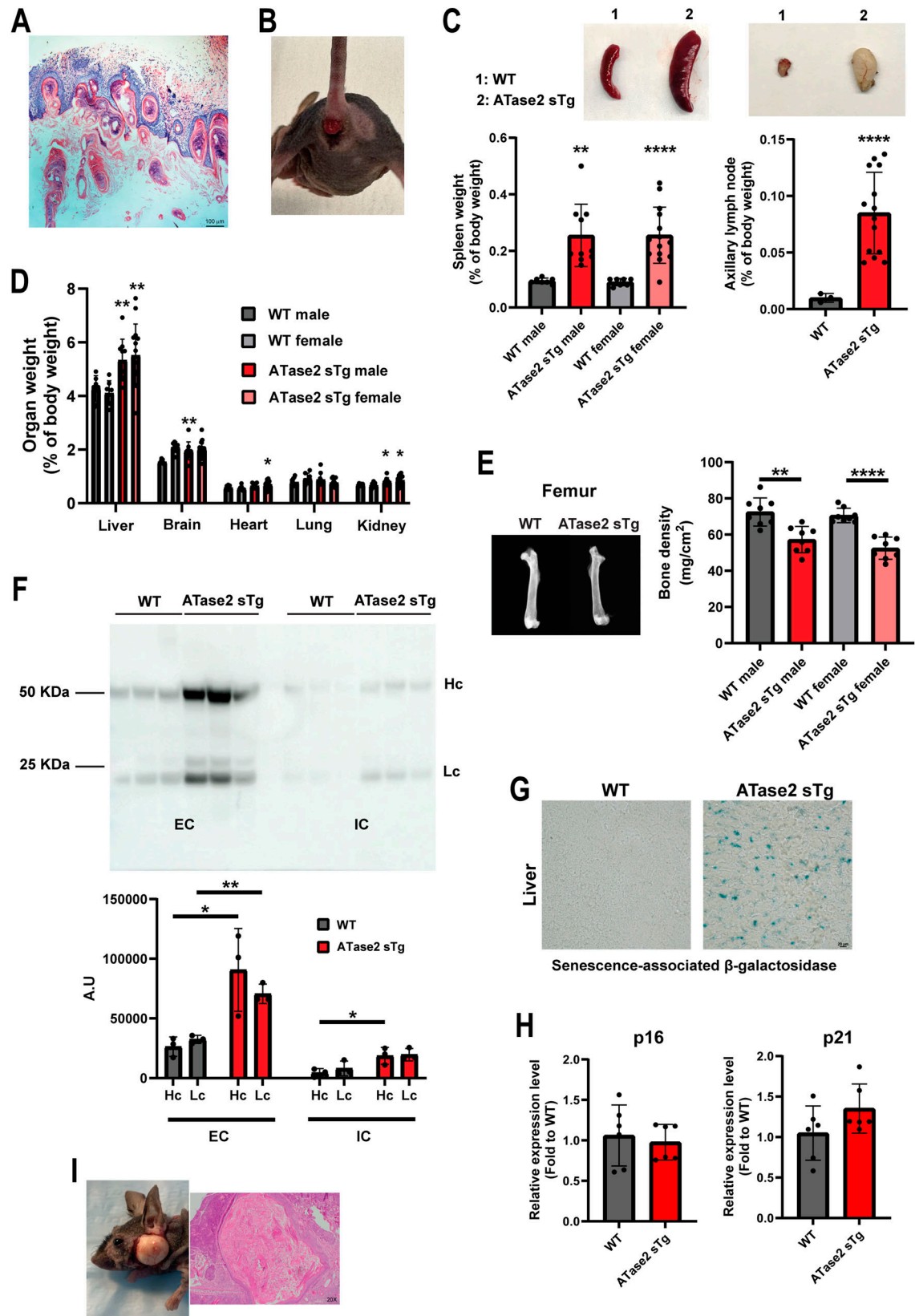

**Figure 4. ATase2 sTg mice displayed progeria-like features.**
**(A)** Representative H&E staining of skin sections of ATase2 sTg mice. **(B)** Rectal prolapse in ATase2 sTg mice. **(C)** ATase2 sTg mice displayed splenomegaly and adenomegaly (axillary lymph nodes are shown). Top panel, representative images. Bottom panel, quantification. The number of animals is shown. **P < 0.005; ****P < 0.0001. Welch't test. **(D)** Organ to body weight ratio. The number of animals is shown. *P < 0.05; **P < 0.005. Unpaired t test or Welch't test. **(E)** ATase2 sTg mice displayed

terminals. These findings clearly implicate the ATases with the engagement and efficiency of the secretory pathway (discussed later).

### ATase1 sTg and ATase2 sTg mice display different engagement of selected pathways

Next, we examined the engagement of selective pathways that appear to differentiate the ATase1 and ATase2 sTg models at the proteomic level. First, we targeted cluster 2 (nucleosome core/histones) in ATase1 sTg, which did not emerge from the Markov clustering of the ATase2 sTg dataset (Fig 5C). Most of ATase1 sTg cluster 2 proteins appeared to be up-regulated, when compared with WT mice (Fig 8A) and were further clustered under four different subcategories with histones and ATP-dependent chromatin remodeling proteins being the most represented (Fig 8B). Only three proteins appeared to be down-regulated (Fig 8A): Brd3, a bromodomain containing protein that recognizes and binds acetylated histones (LeRoy et al, 2008; Daneshvar et al, 2020); Herc2 and Rnf20, which might be involved in the regulation of chromosomal structure (Zhu et al, 2005).

As mentioned above, AT-1 acts as an antiporter by coupling the cytosol-to-ER flux of acetyl-CoA with the ER-to-cytosol flux of free CoA (Fig 8C). The antiporter activity is regulated by the availability of free CoA within the ER lumen, which depends on the acetyl-CoA: lysine acetyltransferase activity of the ATases (Fernandez-Fuente et al, 2023b). Therefore, it is possible that, in addition to changes in protein levels of nuclear/chromatin associated proteins, as reflected in Fig 8A, ATase1 sTg mice might display changes in the acetylation profile of core histone proteins caused by changes in acetyl-CoA availability in the cytosol and nucleus (Fig 8C). Immunoblot assessment revealed reduced acetylation of different histone marks (specifically: K5-H2A, K9-H3, K27-H3, and K56-H3), although only K9-H3 and K27-H3 reached statistical level of significance (Fig 8D). When taken together, the above results point to a marked adaptive nuclear response in ATase1 sTg mice, which was achieved through changes in protein levels and posttranslational modification of core histone proteins although the former mechanism appeared to be the most dominant.

Next, we targeted cluster 2 (ubiquitin/proteasome/ER stress) in ATase2 sTg mice, which did not emerge from the Markov clustering of the ATase1 sTg dataset (Fig 5C). The two major subgroups within the cluster were related to the ubiquitin and proteasome machinery (Fig 9A and B). Indeed, there was a high representation of COP9 signalosome complex subunits, proteasome structural subunits, and different components of the ubiquitin system (Fig 9A and B). One of the biological functions currently associated with the ATase is the induction of autophagy from the ER (Fernandez-Fuente et al, 2023b). This was clearly documented in Atase1$^{-/-}$ and Atase2$^{-/-}$ mice, both displaying increased activation of the autophagy machinery (Rigby et al, 2021). Since a block in macro-autophagy can result in increased activity of the ubiquitin/proteasome system (Komatsu et al, 2005, 2006; Behrends et al, 2010), we tested whether ATase2 sTg mice displayed increased levels of ubiquitin conjugated species. This was indeed the case, as documented by a generalized increase in the levels of ubiquitinated structures (Fig 9C and D).

In conclusion, systemic overexpression of either ATase1 or ATase2 in the mouse causes a similar but not identical progeria-like phenotype. The widespread proteomic alterations observed implicate the ATases with engagement of the secretory pathway, protein homeostasis, and metabolic crosstalk between different cellular compartments.

## Discussion

ATase1 and ATase2 are two ER-based acetyltransferases responsible for Nε-lysine acetylation of ER-cargo and -resident proteins. They work in concert with AT-1, an ER-membrane transporter, which ensures availability of acetyl-CoA into the ER lumen (Fig 10). Cytosolic acetyl-CoA mainly originates from citrate, through the ATP citrate lyase (ACLY), or acetate, through the acetyl-CoA synthetase 2 (ACSS2). The supply of citrate depends on SLC25A1, which exchanges citrate for malate across the mitochondrial membrane, and SLC13A5, which imports citrate and Na$^+$ from the extracellular *milieu* (Fernandez-Fuente et al, 2023b). In the mouse, the ATases respond more dramatically to the citrate/acetyl-CoA flux, as dictated by the SLC13A5/AT-1 network (see Fig 10) (Fernandez-Fuente et al, 2023a). Indeed, systemic overexpression of SLC13A5 leads to up-regulation of endogenous AT-1, increased transport of acetyl-CoA into the ER lumen, and increased ATase-mediated acetylation of ER cargo and transiting proteins (Fernandez-Fuente et al, 2023a). The animals develop a multisystemic phenotype resembling segmental forms of human progerias. Both the hyperacetylation of ER cargo/transiting proteins and the progeria-like phenotype are rescued by the inhibition of the ATases (Fernandez-Fuente et al, 2023a). Neither the up-regulation of AT-1 nor the progeria-like phenotype is observed in mice with systemic overexpression of SLC25A1 (Fernandez-Fuente et al, 2023a). The differential behavior of SLC13A5 and SLC25A1 sTg mice is likely caused by their different transport properties, with the antiporter mechanism of SLC25A1 being limited by the availability of cytosolic malate (Fernandez-Fuente et al, 2023a). Systemic overexpression of AT-1 also leads to hyperacetylation of ER cargo/transiting proteins and a progeria-like phenotype that is rescued by the inhibition of the ATases (Peng et al, 2018; Murie et al, 2022). Finally, mice with systemic overexpression of ATase1 or ATase2 develop a multisystemic phenotype resembling segmental forms of human progerias (present study). Salient features included skin alterations,

---

reduced bone density (Femur is shown). The number of animals is shown. **$P < 0.005$; ****$P < 0.0001$. Unpaired $t$ test. **(F)** ATase2 sTg mice displayed increased Ig tissue liver infiltration. The top panel shows the Western blot while the bottom panel shows the associated quantification. The number of animals is shown. *$P < 0.05$; **$P < 0.005$. Unpaired $t$ test. EC, extracellular; IC, intracellular; Hc, heavy chain; Lc, light chain. **(G)** Senescence associated β-galactosidase staining (liver). **(H)** Senescence markers p16 and p21 (brain). **(I)** Representative gross photograph and H&E-stained section of tumor-like epidermal inclusion cyst in one ATase2 sTg mouse. Mice were ~3 mo old when studied.
Source data are available for this figure.

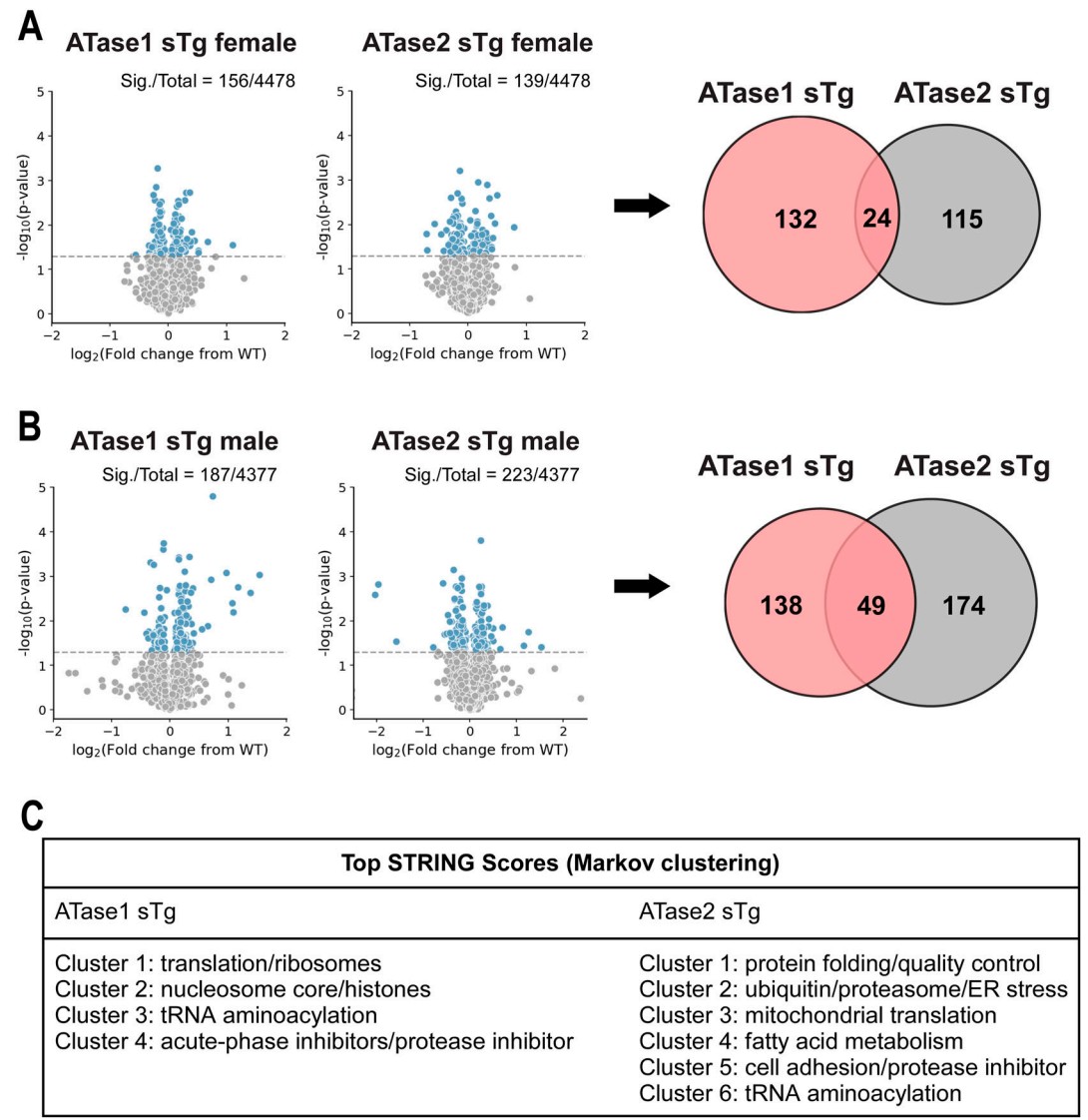

**Figure 5. ATase1 sTg and ATase2 sTg mice displayed significant proteomic changes with minor overlap among models.**
**(A)** Left panel, volcano plot of ATase1 and ATase2 sTg female mice. Significantly changed proteins are shown in blue. Right panel, Venn diagram with overlapping proteins among models. Only significantly altered proteins are shown. N = 4 mice/group. **(B)** Left panel, volcano plot of ATase1 and ATase2 sTg male mice. Significantly changed proteins are shown in blue. Right panel, Venn diagram with overlapping proteins among models. Only significantly altered proteins are shown. N = 4 mice/group. **(C)** Top STRING protein clusters in ATase1 sTg and ATase2 sTg mice. Mice were ~3 mo old when studied.

lordokyphosis, reduced bone density, splenomegaly, adenomegaly, and systemic inflammation (see Table 1). Overall, the phenotype of ATase1 and ATase2 sTg mice resemble the SLC13A5 sTg and AT-1 sTg phenotype (present study and [Peng et al, 2018; Fernandez-Fuente et al, 2023a]). In essence, the ATases appear to be the last output of the SLC13A5/AT-1 metabolic network that is intimately linked to a segmental form of progeria (see Fig 10). Support to this conclusion comes from human-based studies. Indeed, gene duplication events involving 17p13.1 (harboring SLC13A5), 3q25.31 (harboring AT-1), and 2p13.1 (harboring both ATases) are all associated with autism spectrum disorder with intellectual disability and progeria-like dysmorphism (see National Organization for Rare Disorders database; see also [Francke,

1978; Fineman et al, 1983; Fryns et al, 1989; Sawyer et al, 1994; Rizzu et al, 1997; Ounap et al, 2005; Krumm et al, 2013; Poultney et al, 2013; Carvalho et al, 2014; Mooneyham et al, 2014; Krumm et al, 2015]).

Progeroid syndromes are rare genetic disorders that manifest with delayed growth, facial dysmorphism and a complex multi-systemic phenotype that mimics some features of an accelerated/pathogenic form of aging. Although with obvious limitations, they provide valuable models for studying the mechanisms of aging and age-associated diseases. Most studied forms of progerias are characterized by defects of the nuclear envelope or DNA instability. They include Werner syndrome, Hutchinson-Gilford syndrome, Cockaye syndrome, Xeroderma pigmentosum, Rothmund-Thomson syndrome, and Bloom syndrome. Both the genome

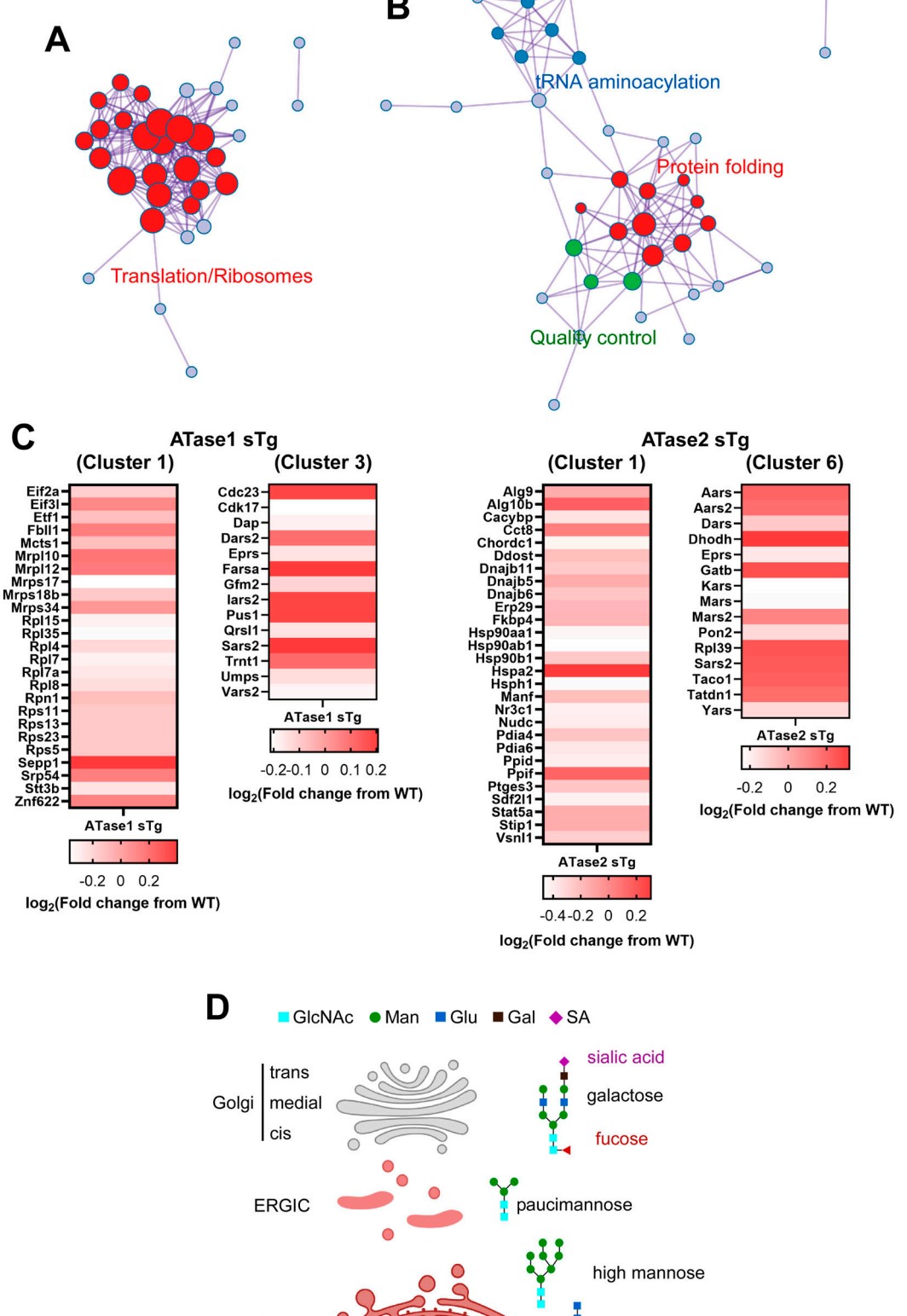

instability and progeroid-like features are efficiently reproduced in the mouse (Bol et al, 1998; Chester et al, 1998; Lebel & Leder, 1998; Pendas et al, 2002; Osorio et al, 2011; Yokoyama et al, 2019). The mechanism underlying the SLC13A5, AT-1, and ATase duplication-associated syndromes differs substantially from the above and offers a unique model where specific metabolic outputs disrupt the proteostatic functions of the ER altering the quality of the secretome (discussed later). The position of the ATases as the last output of the pathway (Fig 10) might also offer mechanistic and therapeutic avenues for several age-associated diseases.

The entire ER acetylation machinery (AT-1 and the two ATases) has emerged as a novel branch of the more general nutrient-signaling pathway that ensures intracellular metabolic connectivity (Fernandez-Fuente et al, 2023b). Specifically, the ER acetylation machinery responds to the availability of acetyl-CoA to influence functional dynamics of the secretory pathway by regulating proteostasis within the ER/secretory pathway (Fernandez-Fuente et al, 2023b). This is clearly reflected by both the proteome and glyco-proteome alterations observed in the brain of ATase1 and ATase2 sTg mice. At the level of the proteome, we found the largest changes within clusters that are broadly involved with the bio-synthesis, folding, N-glycosylation, and quality control of nascent glycoproteins within the ER lumen (see Fig 5). At the level of the glycoproteome, we found significant alterations within post-ER events. This was reflected by changes in the levels of high mannose, fucosylated, and sialylated structures, which clearly point to altered ER-to-Golgi and intra-Golgi transition (see Figs 6D and 7). In essence, the proteome and glycoproteome datasets point to a converging outcome: overexpression of the ATases disrupts the ability of the ER to control the engagement of the secretory pathway and the quality of the *secretome*. On this regard, it is important to stress two concepts: (1) the quality of the *secretome*, which is embedded in the primary sequence of secreted proteins, depends on the ability of the cell to ensure required post-translational modifications and transport of nascent proteins to the right place; (2) N-glycosylation heavily influences protein activity, and defective N-glycosylation is the underlying basis of several congenital diseases (Sturla et al, 2001; Klaric & Lauc, 2022; Pasala et al, 2024; Raynor et al, 2024; Pan & Zhang, 2025). Consistently, the glycoproteome analysis of the ATase sTg models highlighted fundamental biological functions that are intimately connected to neuronal biology and are highly dependent on dynamics of the secretory pathway. They included cell surface-dependent events, cell-cell adhesion/interactions, extracellular matrix organization, assembly and activity of synaptic terminals (see Fig 7). These findings implicate the ATases with the engagement and efficiency of the secretory pathway.

The most notable differences within the proteome of ATase1 and ATase2 sTg mice were found with the nucleosome core/histone and ubiquitin/proteasome/quality control clusters with the former being represented in ATase1 sTg mice and the latter being represented in ATase2 sTg mice (see Fig 5C). Again, these

differences are likely to reflect distinct biological attributes. Interestingly, both Atase1$^{-/-}$ and Atase2$^{-/-}$ mice displayed increased engagement of the autophagy machinery (Rigby et al, 2021). However, only Atase2$^{-/-}$ mice displayed activation of the Atf6 canonical ER stress signaling pathway together with a more evident stimulation of macroautophagy (Rigby et al, 2021). Furthermore, the analysis of the acetyl-proteome of the knock-out models revealed a heavy predominance of proteasome categories in the Atase2$^{-/-}$ model that did not appear in Atase1$^{-/-}$ mice (Rigby et al, 2021). In essence, the knock-out and overexpressing models offer convergent insights into the differential biological functions of the ATases within the ER.

The acetyl-CoA:lysine acetyltransferase activity of the ATases yields free CoA, which can exit the ER through AT-1 (Fig 10) (Fernandez-Fuente et al, 2023b). The antiporter activity of AT-1 is regulated by the concentration gradients of both acetyl-CoA (in the cytosol) and free CoA (in the ER lumen). As a result, the activity of the two ATases can affect the antiporter activity of AT-1 by determining the levels of free CoA within the ER lumen (Rigby et al, 2021; Fernandez-Fuente et al, 2023b). While analyzing Atase1$^{-/-}$ and Atase2$^{-/-}$ mice, we noticed that the excess cytosolic acetyl-CoA accumulating from slowdown of the At-1 antiporter activity was being used differently in the two models: increased acetylation of non-ER/secretory proteins in Atase1$^{-/-}$ mice and accumulation of lipid droplets in Atase2$^{-/-}$ mice. A somewhat similar outcome emerged with the ATase1 sTg mice, which displayed a clear up-regulation of several histone and core nucleosome proteins together with reduced histone acetylation.

Overall, our results link a hyperactive ER acetylation machinery to a segmental form of progeria and reinforce the link between ER acetylation and proteostasis within the secretory pathway. Our study also reinforces the concept of biological divergent roles for ATase1 and ATase2. Finally, ATase1 sTg and ATase2 sTg mice, together with AT-1 sTg mice, offer unique insights on the molecular aspects of progeria and pathogenic forms of aging that differ from nuclear/genomic instability models.

# Materials and Methods

### Transgenic mouse generation

pTRE-ATase1 transgenic and pTRE-ATase2 transgenic mice were generated as described (Hullinger et al, 2016; Rigby et al, 2022a, 2022b). In brief, human cDNA was isolated from ATase1- and ATase2-pCMV6 plasmids (RC215647 for ATase1/NAT8B and RC202157 for ATase2/NAT8; Origene) using PCR and subsequently subcloned into pTRE-Tight plasmid (Takara Bio, Inc.). The resulting pTRE-Tight-ATase1 and pTRE-Tight-ATase2 plasmids were linearized with XhoI and then injected (3 ng/µl) into the pronuclei on one-cell C57BL/6J embryos (Stock No. 000664; The Jackson Laboratory). Rosa26:tTA; pTRE-ATase1 (ATase1 sTg) and Rosa26:tTA; pTRE-ATase2

**Figure 6. ATase1 sTg and ATase2 sTg mice displayed widespread changes among the protein biosynthetic pathway.**
**(A, B)** Ontology enrichment of clusters 1 and 3 from ATase1 sTg (A) and clusters 1 and 6 from ATase2 sTg mice (B). Cluster labeling is shown in Fig 5C. **(C)** Heatmaps of clusters 1 and 3 from ATase1 sTg and cluster 1 and 6 from ATase2 sTg mice. Cluster labeling is shown in Fig 5C. **(D)** Schematic diagram of N-glycosylation across the secretory pathway. Mice were ~3 mo old when studied.

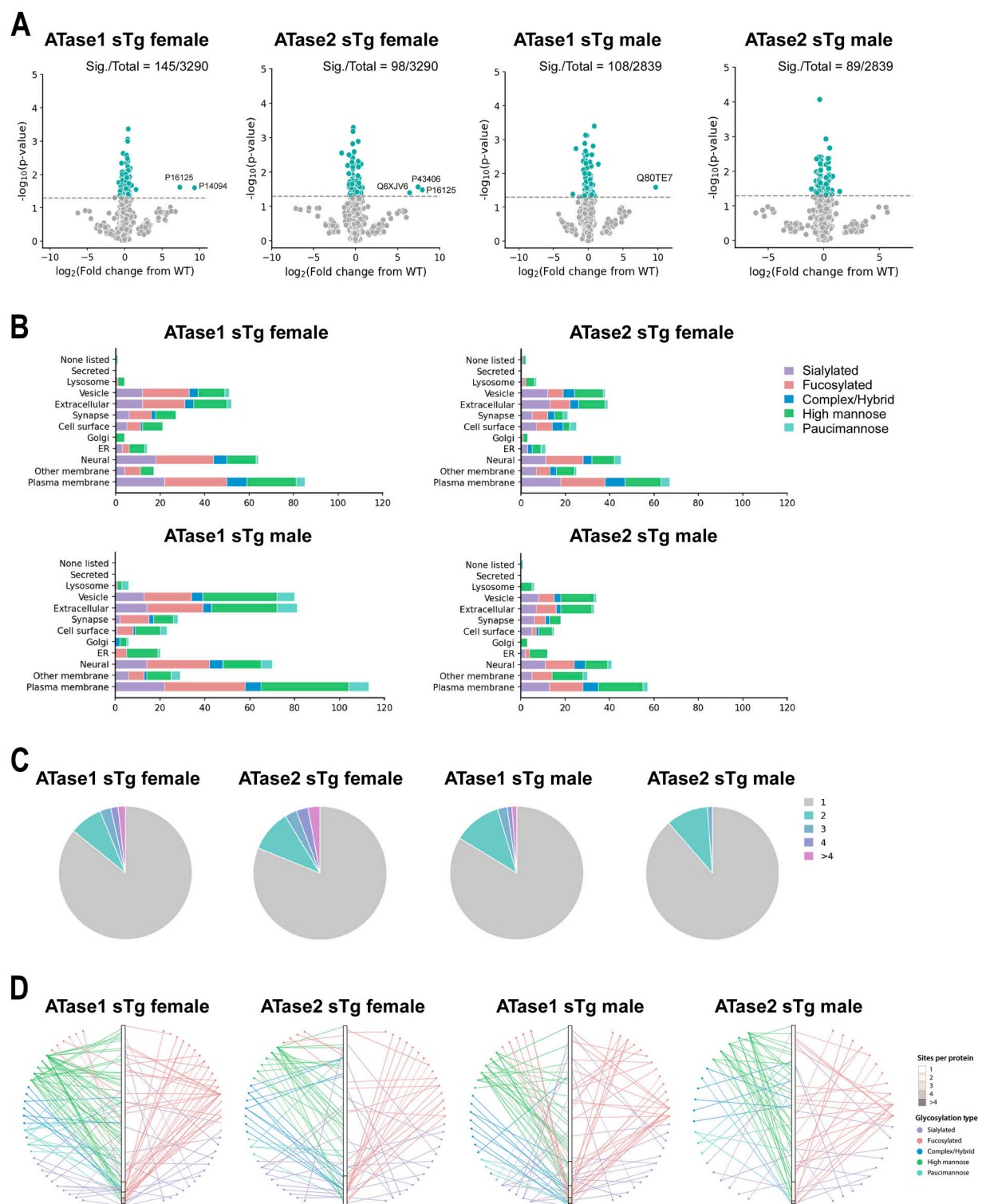

**Figure 7. ATase1 sTg and ATase2 sTg mice displayed significant alterations within the N-glycoproteome (cortex).**
**(A)** Volcano plots showing quantified glycoproteins. Statistically significant glycoproteins are shown in green. N = 4 mice/group. Significance was calculated at *P* < 0.05 via *t* test. **(B)** Glycosylation distribution across different subcellular localizations as defined by GO cellular component terms. Only significantly altered glycans are shown. **(C)** Pie charts showing the distribution of altered glycans per glycosite. Only statistically significant glycans are shown. **(D)** Association network of the type and number of glycans per glycoprotein. Only statistically significant glycans are shown. Mice were ~3 mo old when studied.

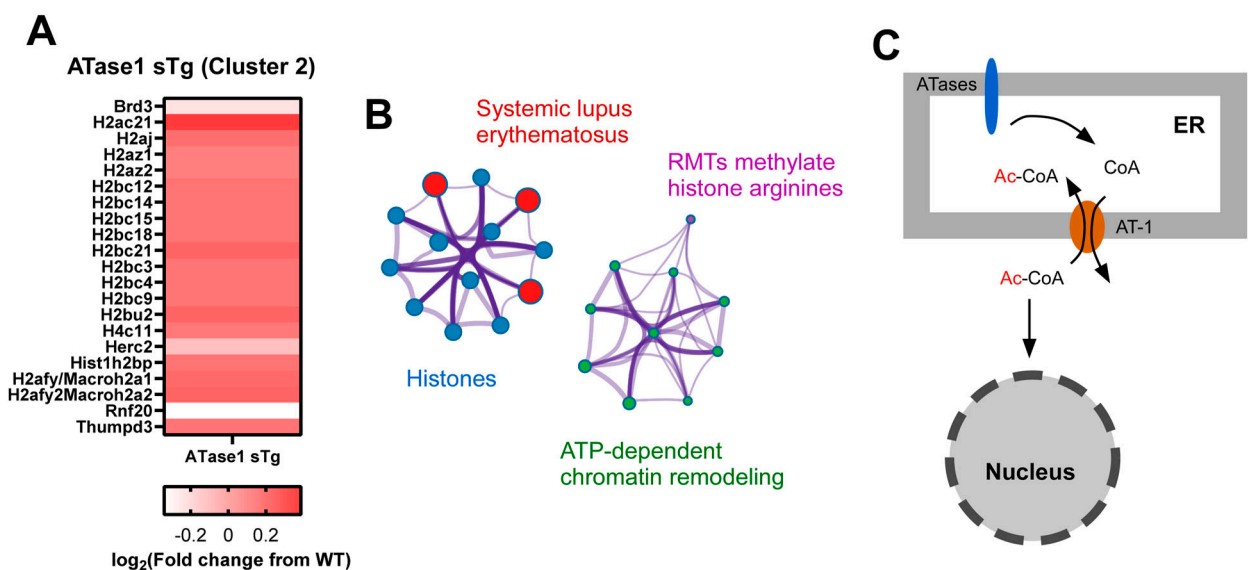

**A** ATase1 sTg (Cluster 2)

Brd3
H2ac21
H2aj
H2az1
H2az2
H2bc12
H2bc14
H2bc15
H2bc18
H2bc21
H2bc3
H2bc4
H2bc9
H2bu2
H4c11
Herc2
Hist1h2bp
H2afy/Macroh2a1
H2afy2Macroh2a2
Rnf20
Thumpd3

ATase1 sTg

−0.2  0  0.2

log₂(Fold change from WT)

**B**
Systemic lupus erythematosus
RMTs methylate histone arginines
Histones
ATP-dependent chromatin remodeling

**C**
ATases
ER
Ac-CoA → CoA
AT-1
Ac-CoA
Nucleus

**D**

(Western blots: K5-H2A, K5-H2B, K9-H3, K14-H3, K18-H3, K27-H3, K56-H3, K5-H4, K8-H4, K12-H4, H2A, H2B, H3, H4 for WT and ATase1 sTg)

Bar graph: Relative intensity (normalize to WT)
K5-H2A, K5-H2B, K9-H3 **, K14-H3, K18-H3, K27-H3 *, K56-H3, K5-H4, K8-H4, K12-H4, H2A, H2B, H3, H4 p=0.05

**Figure 8. Nuclear adaptive response in ATase1 sTg mice.**
**(A, B)** Heatmap (A) and ontology enrichment (B) of cluster 2 from ATase1 sTg mice. Cluster labeling is shown in Fig 5C. **(C)** Changes in ATase activity in the ER lumen can affect the availability of acetyl-CoA in the cytosol and nucleus by regulating the antiporter activity of AT-1 (see also Fig 10). **(D)** Western blot and associated quantitation

(ATase2 sTg) were generated by crossing Rosa26:tTA mice with pTRE-ATase1 and pTRE-ATase2 mice, respectively. Genotyping from tail DNA was performed by TransnetYX using the following primers: Rosa26:tTA forward (5′-GCCGTGGGCCACTTCA-3′), Rosa26:tTA reverse (5′-CTGGTGCTCCTGGTCCTC-3′), pTRE-ATase1 forward (5′- GCT CGTTTAGTGAACCGTCAGAT-3′), pTRE-ATase1 reverse (5′- CTCCTGGTA TTTGCGGATGTGAT-3′), pTRE-ATase2 forward (5′- GCTCGTTTAGTGAAC CGTCAGAT-3′), and pTRE-ATase2 reverse (5′- CTCCTGGTATTTGCG GATGTGA-3′).

### Animals

All experimental animals were housed in the standardized cages provided by the University Laboratory Animal Resources and grouped with 1–5 mice per cage. Mice were supplied with either standard diets or diets with doxycycline (200 mg/kg) purchased from Bio-Serv and water ad libitum. All animal experiments were performed in accordance with the National Institutes of Health Guide for the Care and Use of Laboratory Animals and were approved by the Institutional Animal Care and Use Committee of the University of Wisconsin-Madison (protocol #M005120). Non-transgenic, WT C57BL/6J mice (Stock No. 000664; The Jackson Laboratory) were used as controls (WT). Age, sex, and number of animals used with each experiment are reported in the figure legends.

### Western blotting

Western blotting was conducted as previously described (Peng et al, 2016, 2018). The following primary antibodies were used in this study: $\beta$-actin (#3700, 1:5,000; Cell Signaling; #4967, 1:5,000; Cell Signaling), ATase1/NAT8B (#PA5-77166, 1:1,000; Invitrogen), ATase2/NAT8 (#NBP1-47863, 1:1,000; NOVUS), Acetyl-Histone H2A Lys5 (#2576, 1:1,000; Cell Signaling), Acetyl-Histone H2B Lys5 (#12799, 1:1,000; Cell Signaling), Acetyl-Histone H3 Lys9 (#9649, 1:1,000; Cell Signaling), Lys14 (#7627, 1:1,000; Cell Signaling), Lys18 (#13998, 1:1,000; Cell Signaling), Lys27 (#8173, 1:1,000; Cell Signaling), Lys56 (#4243, 1:1,000; Cell Signaling), Acetyl-Histone H4 Lys5 (#8647, 1:1,000; Cell Signaling), Lys8 (#2594, 1:1,000; Cell Signaling), Lys12 (#13944, 1:1,000; Cell Signaling), Histone H2A (#12349, 1:1,000; Cell Signaling), Histone H2B (#12364, 1:1,000; Cell Signaling), Histone H3 (#4499, 1:1,000; Cell Signaling), Histone H4 (#13919, 1:1,000; Cell Signaling), and Ubiquitin (#3936, 1:1,000; Cell Signaling). Both donkey anti-rabbit and goat anti mouse IRDye 800CW and 680RD-conjugated secondary antibodies (#926-32213; #926-68070; #926-32210; #926-68073; LI-COR) were used for infrared imaging (LI-COR Odyssey Infrared Imaging System; LI-COR Biosciences). For immunoglobin measurement, extracellular and intracellular fraction were prepared as previously described (Peng et al, 2018; Fernandez-Fuente et al, 2023a). In brief, liver tissue was homogenized in extracellular buffer (50 mM Tris–HCl, pH 7.6; 150 mM NaCl; 2 mM EDTA; 0.01 [wt/vol] SDS; and 0.01% [vol/vol] NP-40) with

inhibitors. The supernatant was collected as extracellular fraction after centrifugation for 90 min at 17,142$g$ with Eppendorf rotor FA-45-24-11-HS at 4°C. Pellet was then homogenized in intracellular buffer (50 mM Tris–HCl, pH 7.6; 150 mM NaCl; 2 mM EDTA and 0.01 [wt/vol] Triton X-100) with inhibitors. The supernatant was collected as intracellular fraction after centrifugation for 90 min at 17,142$g$ with Eppendorf rotor FA-45-24-11-HS at 4°C. For nuclear fraction, target signals were normalized to total protein using Revert Total Protein Stain (#926-11021; LI-COR). Uncropped Western blot images are shown in the Supporting Data section.

### Bone density

Femur and Tibia were collected and fixed in 70% ethanol. Bone density was measured by the UltraFoxus DXA system (Faxitron) using the standard manufacturer protocols.

### Reverse transcription quantitative PCR (RT-qPCR)

Liver tissue was used for RNA extraction (#R1054; Zymo), cDNA synthesis (#18080400; Invirogen) and quantitative PCR using Roche 480 LightCycler and SYBR Green I Master (#04887352001; Roche). Expression levels were normalized to $\beta$-actin. Relative expression levels were normalized to control and expressed as fold change. The following primers were used: $\beta$-actin forward (5′-CTAAGGCCA ACCGTGAAAAG-3′), $\beta$-actin reverse (5′-ACCAGAGGCATACAGGGACA-3′), p16 forward (5′-GTGTGCATGACGTGCGGG-3′), p16 reverse (5′-GCA GTTCGAATCTGCACCGTAG-3′), p21 forward (5′-AACATCTCAGGGCCG AAA-3′), p21 reverse (5′-TGCGCTTGGAGTGATAGAAA-3′).

### Senescence $\beta$-galactosidase assay

Senescence $\beta$-Galactosidase was detected by using Senescence $\beta$-galactosidase Staining Kit (#9860S; Cell signaling Technology). In brief, mouse liver cryosections (10 $\mu$m) were fixed and stained with $\beta$-galactosidase at 37°C overnight.

### Histology

Samples for H&E staining were immediately fixed with 10% neutral buffered formalin overnight after collection. Samples were then embedded in paraffin by standard techniques and section on microtome. After deparaffinization and rehydration, sections were stained with H&E staining.

### 12-plex DiLeu labeling

12-plex dimethylated leucine (DiLeu) labeling was performed as previously described (Frost et al, 2015; Dieterich et al, 2021). Briefly, dissected brain samples of ATase1 sTg, ATase2 sTg, and WT littermates were homogenized and lysed in 8 M urea buffer with a probe sonicator. The extracted proteins were reduced by 5 mM DTT

showing the histone acetylation profile of ATase1 sTg mice (brain cortex nuclear fraction; age: 6 mo). The number of animals is shown. *$P < 0.05$; **$P < 0.005$. Unpaired $t$ test. Mice were ~6 mo old when studied.
Source data are available for this figure.

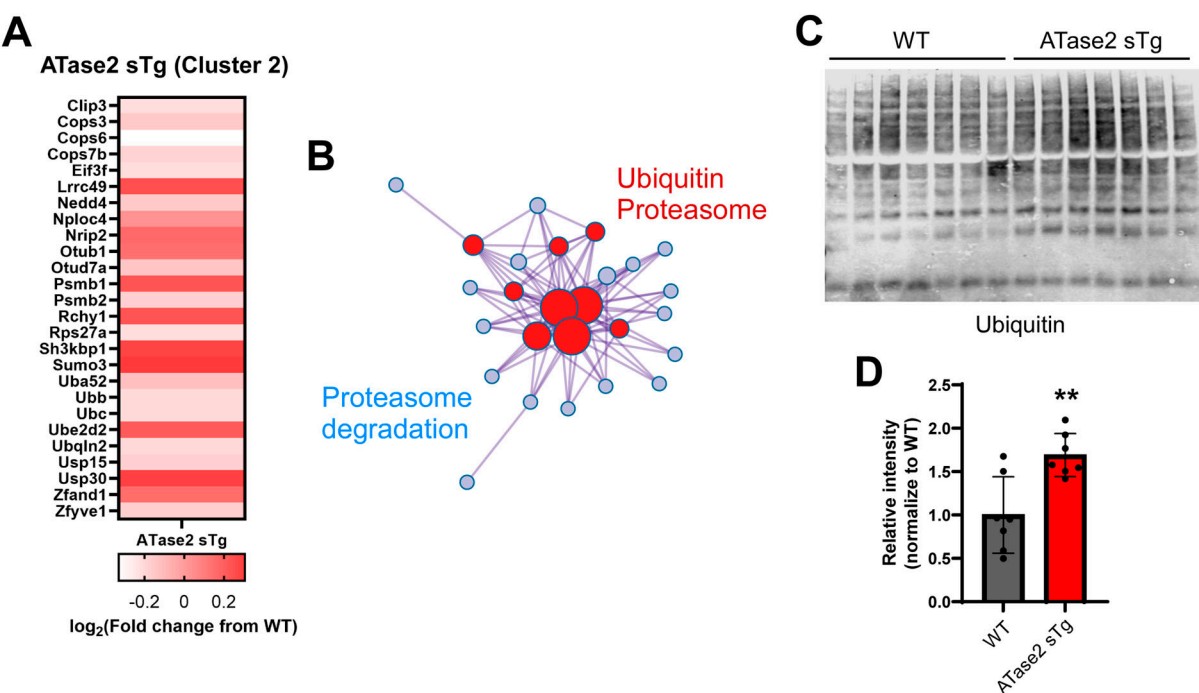

**Figure 9. Adaptive response within the ubiquitin pathway in ATase2 sTg mice.**
**(A, B)** Heatmap (A) and ontology enrichment (B) of cluster 2 from ATase2 sTg mice. Cluster labeling is shown in Fig 5C. **(C, D)** Western blot (C) and associated quantitation (D) showing increased ubiquitination in ATase2 sTg (brain cortex; age: 3 mo). The number of animals is shown. **$P < 0.005$. Unpaired $t$ test. Mice were ~3 mo old when studied.
Source data are available for this figure.

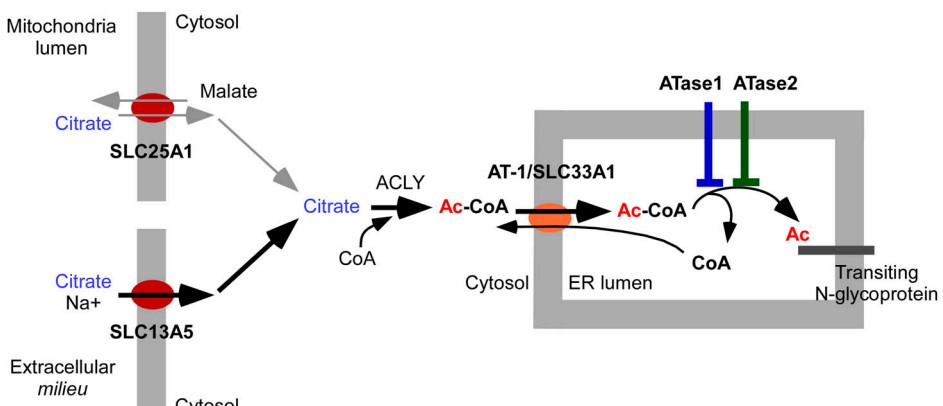

**Figure 10. Schematic view of the citrate/acetyl-CoA pathway with the ER acetylation machinery.**
Description with relevant references is in the text. The ER acetylation machinery responds more dramatically to the citrate imported by SLC13A5 (black arrows) than SLC25A1 (gray arrows).

at RT for 1 h, followed by alkylation with 15 mM IAA for 30 min in the dark. The alkylation reaction was quenched by adding DTT to a final concentration of 5 mM. The alkylated proteins were then diluted and digested with trypsin at an enzyme-to-protein ratio of 1:50 at 37°C for 18 h. The resulting tryptic peptides were desalted using C18 SepPak cartridges, dried under vacuum, and reconstituted in 0.5 M TEAB for labeling. DiLeu tags were dissolved in anhydrous DMF and activated with DMTMM and NMM at a 0.6× molar ratio relative to the tags. The mixture was vortexed at RT for 1 h. The reaction mixture was vortexed at RT for 1 h. After centrifugation, the supernatant was immediately mixed with tryptic peptides from a single condition at a 10:1 tag-to-peptide (wt/wt) ratio and incubated with vortexing at RT for 2 h. The reaction was quenched by NH$_2$OH. Labeled peptide batches were combined to generate 12-plex mixtures. An aliquot of the dried peptides was further processed using SCX spin tips according to the manufacturer's protocol.

**High pH fractionation**

High-pH (HpH) fractionation was conducted using a Waters Alliance e2695 HPLC equipped with a C18 reversed-phase column (2.1 ×

150 mm$^2$, 5 $\mu$m, 100 Å) at a flow rate of 0.2 ml/min. Mobile phase A consisted of 10 mM ammonium formate at pH 10 adjusted with ammonium hydroxide while mobile phase B consisted of 90% ACN and 10 mM ammonium formate at pH 10. Separation was achieved using the following gradient: 1% B (0–5 min), 1–40% B (5–50 min), 40–60% B (50–54 min), 60–70% B (54–58 min), and 70–100% B (58–59 min). Fractions were collected every 2 min, and nonadjacent fractions were pooled into eight groups before being dried under vacuum for subsequent LC-MS/MS analysis.

### Glycopeptide enrichment

DiLeu labeled glycopeptides were enriched using in-house packed strong anion-exchange (SAX)-HILIC SPE tips following a previously reported protocol with minor modifications (Selman et al, 2011). 3 mg of cotton wool was inserted into an empty TopTip. PolySAX LP bulk material was prepared as a 10 mg/200 $\mu$l slurry in 1% TFA and activated by vigorous vortexing for 15 min. The activated slurry was then transferred to the spin tip at a beads-to-peptide mass ratio of 30:1. Solvent removal was performed by centrifugation at 121$g$ with Eppendorf rotor F45-24-11 for 2 min, ensuring the SAX material was firmly packed at the top of the tip. The stationary phase was conditioned with 300 $\mu$l of 1% TFA and 300 $\mu$l of loading buffer (80% ACN, 1% TFA), with each step repeated three times. DiLeu-labeled peptides were aliquoted to a total of 1,800 $\mu$g, with each aliquot dissolved in 300 $\mu$l of loading buffer before loading onto the tips by centrifugation at 121$g$ with Eppendorf rotor F45-24-11 for 2 min. The flow-through was collected and reloaded to ensure complete peptide retention. The tips were then washed six times with 300 $\mu$l of loading buffer. Elution was performed sequentially in four fractions using 300 $\mu$l of 70% ACN with 0.1% FA, 50% ACN with 0.1% FA, 25% ACN with 0.1% FA, and 0% ACN with 0.1% FA. Each eluted fraction was collected separately and dried under vacuum before MS analysis.

### LC–MS/MS analysis

Enriched glycopeptides from each fraction were reconstituted in 0.1% FA and analyzed by reversed-phase LC-MS/MS with an Orbitrap Fusion Lumus coupled to a Dionex UltiMate 3000 UPLC system. Peptides separation was performed on an 18 cm × 75 $\mu$m i.d. custom-packed BEH C18 (1.7 $\mu$m, 130 Å) capillary column using an 80-min gradient from 0% to 30% ACN containing 0.1% FA. Data acquisition was carried out in top-speed mode with a 3-s cycle time. Precursor ion scans were recorded over an m/z range of 400–2,000 at a resolution of 60,000, with a normalized automatic gain control (AGC) target of 60% and a maximum injection time (IT) of 200 ms. Selected precursor ions underwent higher-energy C-trap dissociation (HCD) with a normalized collision energy (NCE) of 30, incorporating a ±3% stepped HCD collision energy. Tandem MS spectra were acquired at a resolution of 60,000 with a lower mass limit of m/z 110. A dynamic exclusion of 12 s with a 10 ppm mass tolerance was applied.

Peptides from each fraction were reconstituted in 0.1% FA and analyzed using a Dionex UltiMate 3000 UPLC coupled to a Q Exactive HF Orbitrap mass spectrometer. Peptides separation was performed using the same column as described above, with a 100-min gradient from 0 to 30% ACN containing 0.1% FA. The MS scan range covered m/z 300–1,500, also at a resolution of 60,000, with an AGC target of 1E6 and a maximum IT of 100 ms. The MS/MS method used a top 20 data-dependent acquisition (DDA) mode, with all MS/MS dissociations conducted using an NCE of 30. The MS/MS parameters included a resolution of 60,000, an AGC target of 1E5, and a maximum IT of 200 ms. A dynamic exclusion of 45 s with an isolation window of 1.0 m/z was applied.

### Glycoproteomics and proteomics data analysis

Raw files were processed using the Byonic search engine integrated within Proteome Discoverer 2.5. Spectra were searched against the SwissProt *Mus musculus* proteome database (12 January 2021). Trypsin digestion was allowed with up to two missed cleavages. The parent mass tolerance was set to 10 ppm, whereas the fragment mass tolerance was 0.01 D. Fixed modifications included as carbamidomethylation (+57.02146 D) on C residues and 12-plex DiLeu (+145.12801 D) on peptide N-terminus and K. For N-glycoproteomics, dynamic modifications included oxidation of M (+15.99492 D), deamidation (+0.984016 D) of N or Q, and N-glycosylation. Glycan modifications were specified as Byonic embedded mammalian N-glycan database (309 entries). Protein identifications were filtered to a 1% false discovery rate (FDR). Glycopeptides were categorized into five glycan types based on their composition: (1) sialic acid (containing NeuAc/NeuGc), (2) fucose (containing Fucose), (3) complex/hybrid (>2 NeuAc), (4) high-mannose (2 NeuAc and >5 Hex), and (5) paucimannose (2 NeuAc and <5 Hex). Peptide identification results were filtered at Byonic score >100, PEP 2D < 0.05, and |Log Prob| > 1. For proteomics, fixed and dynamic modifications are the same as N-glycoproteomics, the same fixed and dynamic modifications as in N-glycoproteomics were applied, except for the exclusion of N-glycosylation as a dynamic modification. Only high-confidence protein identifications were used for further analysis. Reporter ion intensities from the 12-plex DiLeu channels were median-normalized in Perseus to correct for systematic biases (Tyanova et al, 2016). Gene ontology annotation and *t* test of quantitation results were performed using Perseus (Tyanova et al, 2016). Further data processing was conducted using custom Python and R scripts. Outputs were analyzed with Metascape (Zhou et al, 2019). The LC-MS/MS raw data and annotated spectra have been deposited to the Proteome Xchange Consortium via the MassIVE partner repository with the accession number "MSV000097367".

### Nuclear enrichment

Nuclear fractions were prepared by using EpiQuik Nuclear Extraction Kit (#OP-0002-1; EpiGentek). Briefly, Mouse cortex was homogenized in NE1 buffer containing 0.1% DTT and centrifuged for 10 min at 14,167$g$ with Eppendorf rotor FA-45-24-11-HS at 4°C. The pellet was homogenized in NE2 buffer containing 0.1% DTT and PIC by vortex and sonication, and further centrifuged for 10 min at 19,283$g$ with Eppendorf rotor FA-45-24-11-HS at 4°C. Finally, the supernatant was collected as nuclear fraction for further experiments.

## Statistics

Specific sample size was described in each figure legend. GraphPad Prism 10.0.2 was used to analyze all experimental data. Data were presented as mean ± SD with all the individual data points. Unpaired $t$ test or Welch'$t$ test were used to compare the statistical difference between two groups. Statistical significance was presented as following: *$P < 0.05$, **$P < 0.005$, ***$P < 0.0005$, ****$P < 0.0001$.

# Data Availability

The LC-MS/MS raw data and annotated spectra have been deposited to the ProteomeXchange Consortium via the MassIVE partner repository with the accession number "MSV000097367." Source data for the graphs and charts are available as an excel file labeled as "Data points" and any remaining information can be obtained from the corresponding author upon reasonable request.

# Supplementary Information

# Acknowledgements

The Puglielli laboratory is supported by the National Institute of Health (R01NS094154, R01GM148487, and R01AG078794). The Li laboratory is supported by the National Institute of Health (R01AG052324, P41GM108538, R01AG078794, and R01DK071801). This research also benefitted from a core grant to the Waisman Center from the National Institute of Child Health and Human Development (U54 HD105353). L Li also acknowledges the funding support of NIH shared instrument grants (NIH-NCRR S10RR029531, S10OD028473, and S10OD025084), and a Research Forward grant provided by the University of Wisconsin—Madison Office of the Vice Chancellor for Research with funding from the Wisconsin Alumni Research Foundation. We are also grateful to the members of the Puglielli laboratory for their evaluation of an early version of this manuscript.

## Author Contributions

T-L Cheng: conceptualization, data curation, formal analysis, investigation, visualization, methodology, and writing – original draft, review, and editing.
F Wu: data curation, formal analysis, investigation, visualization, methodology, and writing – original draft, review, and editing.
ME Haque: data curation, formal analysis, investigation, visualization, methodology, and writing – review and editing.
AR Thiel: data curation, formal analysis, investigation, visualization, methodology, and writing – review and editing.
D Wang: data curation, formal analysis, investigation, visualization, methodology, and writing – review and editing.
JJ Helgager: data curation, formal analysis, investigation, visualization, methodology, and writing – review and editing.
L Li: formal analysis, supervision, and writing – review and editing.
L Puglielli: conceptualization, formal analysis, supervision, funding acquisition, project administration, and writing – original draft, review, and editing.

## Conflict of Interest Statement

L Puglielli is a consultant for Belharra Therapeutics. The remaining authors have no competing interests to disclose.

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
