## [Reviewer comments · Life Science Alliance]

Life Science Alliance

Overexpression of ATase1 and ATase2 disrupts the secretome and causes a progeria phenotype

Luigi Puglielli, Tzu-Lin Cheng, Feixuan Wu, Md Ezazul Haque, Abigail Thiel, Danqing Wang, Jeffrey Helgager, and Lingjun Li
DOI: <https://doi.org/10.26508/lsa.202503378>

Corresponding author(s): Luigi Puglielli, University of Wisconsin-Madison and Lingjun Li, University of Wisconsin-Madison

Review Timeline:

Submission Date:	2025-05-02
Editorial Decision:	2025-07-01
Revision Received:	2025-07-31
Editorial Decision:	2025-08-20
Revision Received:	2025-08-27
Accepted:	2025-08-28

Scientific Editor: Tim Fessenden

Transaction Report:

July 1, 2025

Re: Life Science Alliance manuscript #LSA-2025-03378-T

Prof. Luigi Puglielli
University of Wisconsin-Madison
Medicine
Waisman Center
1500 Highland Avenue
Madison, Wisconsin 53705

Dear Dr. Puglielli,

Thank you for submitting your manuscript entitled "Overexpression of ATase1 and ATase2 in the mouse disrupts the quality of the secretome and causes a progeria phenotype" to Life Science Alliance. The manuscript was assessed by expert reviewers, whose comments are appended to this letter. As you will see, all reviewers appreciated these findings on global protein secretion changes due to dysregulated ATases, and the impact these have on development and longevity. We concur with Reviewer 1 that proteomics results should be validated by western blotting for key proteins of interest (point 2). Please also consider the helpful suggestions by this and the other reviewers towards improving this work.

Thank you for this interesting contribution to Life Science Alliance. We are looking forward to receiving your revised manuscript.

Sincerely,

-- By submitting a revision, you attest that you are aware of our payment policies found here: <https://www.life-science->

B. MANUSCRIPT ORGANIZATION AND FORMATTING:

Reviewer #1 (Comments to the Authors (Required)):

There is an interesting submission presenting diverse range of properties of human ATase1-Tg and ATase2-Tg mice as progeria experimental models. Large number of omic-and phenotypic parameters well describes properties and discrete differences between these two animal strains.

Some additions described below could make results and discussion more compatible and thesis more complete.

Introduction. There are some reference background data on acetyl-CoA in ER, that should be taken into consideration

Methods: Age of Tg-asation, and mode of plasmid injection should be given in details.

Results: Fig.8 should accommodate direct measurements of acetyl-CoA, or some appropriate past data ought to be presented in discussion, as the acetyl-CoA and acetylations are key points of that theory.(Fig. 10).

Discussion. Exogenous citrate seems to be less important than the endogenous one as a substrate for ACLY, and thereby as precursor for cytoplasmic acetyl-CoA. Capacity of SLC13A5 is relatively low and [extracellular citrate/Km citrate] ratio is unfavorable vs mitochondrial citrate synthesis and out of mitochondria efflux rates. Thus, endogenous citrate is prevalent.

Significant fraction of acetyl-CoA may be also directly transferred during functional depolarization.

More referrals to figures (absent figs. 1-5) would make discussion more clear. Fig. 10 should be supplemented with endogenous citrate pathway.

Submission is publishable after corrections.

Reviewer #2 (Comments to the Authors (Required)):

Lysine acetylation is one form of post-translational protein modification. In this manuscript, the authors generated transgenic mice that over-express two key enzymes for such modification. As expected, the elevated ATases levels cause serious developmental defects and some lethality, phenotype of which mimics progeria. They noted that expression of ATase2 led to more severe defects, suggesting some difference in substrate specificity when compared to that of ATase1. Finally, they performed detailed proteomic analysis for the secretome of these mice. The results provide some insight into the functions of these ATases. Overall, the findings are of some interests to the audience of LSA. I recommend experimental verification of the proteomic results and additional mechanistic discussion.

1. The changes in protein abundance or Lysine acetylation need to be verified individually, possibly by western blotting or other biochemical approaches. Representative proteins can be selected based on the proteomic analysis. It would be interesting to show different substrate proteins have different changes in Lysine modification upon over-expression of two different ATases.
2. Because protein secretion is the emphasis of the work, it is recommended that levels of COPII components to be compared between the WT and ATase-overexpressed cells or tissues. Similarly, UPR or ERAD activation needs to be checked under these conditions, if it has not been done previously.
3. The key mechanism for which over-expression of ATases disrupts the quality of the secretome needs a little more discussion. Is it a direct change in the the quality control pathway, meaning the pathway being the substrate of these enzymes or the overall accumulation of other inappropriately modified clients?

Reviewer #3 (Comments to the Authors (Required)):

The study implicates ATase 1 and ATase 2 in progeria phenotype. An excellent study done immaculately, with proper stats invoked. The study is slated for even better journals with clinical extrapolation.

Point-by-Point Response

We wish to thank the Editor and the Reviewers for their positive comments. A comprehensive point-by-point response to the suggestions and questions can be found below.

Reviewer #1

Introduction. There are some reference background data on acetyl-CoA in ER, that should be taken into consideration.

Response: Done as requested.

Methods: Age of Tg-asation, and mode of plasmid injection should be given in details.

Response: Done as requested.

Results: Fig.8 should accommodate direct measurements of acetyl-CoA, or some appropriate past data ought to be presented in discussion, as the acetyl-CoA and acetylations are key points of that theory (Fig. 10).

Response: Past data are cited in the Discussion section.

Discussion. Exogenous citrate seems to be less important than the endogenous one as a substrate for ACLY, and thereby as precursor for cytoplasmic acetyl-CoA. Capacity of SLC13A5 is relatively low and [extracellular citrate/Km citrate] ratio is unfavorable vs mitochondrial citrate synthesis and out of mitochondria efflux rates. Thus, endogenous citrate is prevalent. Significant fraction of acetyl-CoA may be also directly transferred during functional depolarization.

Response: We included SLC25A1 in Fig 10. We also expanded the appropriate section in the Discussion and cited our previous work with mice overexpressing SLC25A1 and SLC13A5 (see *Commun Biol.* 2023;6(1):926. doi: 10.1038/s42003-023-05311-1. PMID: PMC10492862). We also cited our recent review where we discussed the biochemistry of SLC25A1 and SLC13A5 (see *Mol Metab.* 2023;67:101653. doi: 10.1016/j.molmet.2022.101653. PMID: PMC9792894).

More referrals to figures (absent figs. 1-5) would make discussion more clear. Fig. 10 should be supplemented with endogenous citrate pathway.

Response: Done as requested. For Fig. 1-4 (phenotype), we only cited Table 1, which summarizes the phenotype.

Reviewer #2

I recommend experimental verification of the proteomic results and additional mechanistic discussion.

Response: For the experimental verification of the proteomic results, please see point 1 below. Additional mechanistic details were added in the Discussion section.

1. *The changes in protein abundance or Lysine acetylation need to be verified individually,*

possibly by western blotting or other biochemical approaches. Representative proteins can be selected based on the proteomic analysis.

Response: LC-MS/MS is a high-sensitivity approach while Western blotting is a low-sensitivity approach. As such, we typically do not perform Western blotting to confirm the LC-MS/MS data. However, to accommodate the request of the reviewer, we targeted 3 representative proteins selected from the proteomic analysis. The results are shown below in **Figure A** and are consistent with the proteomic data. As for lysine acetylation, see the point below with Figure B.

Figure A. Western blot of selected proteomic results. We only tested ATase1 sTg mice because we had frozen tissue available (from mice with the appropriate age).

It is also important to stress that many of the experiments included in the results section were explicitly designed to confirm/verify the results of the proteomic data by using different biochemical approaches. Specifically:

(i) The largest proteomic changes in both sTg models were found to be relevant for protein biosynthesis and regulation of the secretory pathway subgroups (see Fig. 5). The results highlighted changes in the levels of Stt3b, Rpn1 and Ddost, which are integral components of the mouse Ost super-complex, as well as Alg9 and Alg10b, which are involved in the assembly of the initial GlcNAc₂Man₉Glc₃ oligosaccharide structure. To test this possibility, we performed the studies reported in Figure 7, S4, S5, and S6 using HILIC/LC-MS/MS.

(ii) The proteomic data suggested that the nucleosome core/histones cluster was specifically affected in ATase1 sTg (but not ATase2) mice. This was tested with the studies reported in Figure 8 using Western blotting.

(iii) The proteomic data suggested that the ubiquitin/proteasome/ER stress cluster was specifically affected in ATase2 sTg mice. This was tested with the studies reported in Figure 9 using Western blotting.

It would be interesting to show different substrate proteins have different changes in Lysine modification upon over-expression of two different ATases.

Response: The acetylation of individual lysine residues, peptides and proteins is essentially governed by the availability of acetyl-CoA and the kinetics of the individual acetyltransferases. Changes in acetyl-CoA availability as well as changes in the expression levels of the two ATases within the ER lumen are expected to affect both the kinetics (rate) and stoichiometry (extent) of lysine acetylation. Sites with *low-affinity* (low stoichiometry/occupancy) are particularly sensitive and are likely to show greater changes than *high-affinity* (high stoichiometry/occupancy) sites (for example, see: PMID: PMC4118097; PMID: PMC7820588; PMID: PMC6718414; PMID: PMC10492862; MCID: PMC8041774; PMID: PMC9014753; PMID: PMC8823335). In light of the above, we can predict that it would be much easier to detect changes in the stoichiometry of acetylation of individual lysine residues and proteins under situations where one of the ATases is missing (KO) rather than overexpressed (Tg). We have already performed this study, and the results are reported in PMID: PMC8041774.

However, to accommodate the request of the Reviewer, we have performed a crude Western blot with enriched ER from ATase2 sTg mice. The results show increased acetylation of ER proteins (see below **Figure B**). Due to the different behavior of low- and high-affinity sites, we do not plan to look at the stoichiometry of acetylation of these mice.

Figure B. ATase2 sTg mice display increased ER acetylation. Experimental setup: (i) Liver from 3 independent ATase2 sTg mice and 3 independent littermates. (ii) We processed the tissue to obtain enriched ER (using a standardized protocol). (iii) We run a Western blot probing with an anti-acetylated lysine antibody (left panel). The total protein labeling is also included in the figure (right panel).

NOTE: we only tested ATase2 sTg mice because we had intact (frozen) liver tissue (from mice with the appropriate age).

2. Because protein secretion is the emphasis of the work, it is recommended that levels of COPII components to be compared between the WT and ATase-overexpressed cells or tissues. Similarly, UPR or ERAD activation needs to be checked under these conditions, if it has not been done previously.

3. The key mechanism for which over-expression of ATases disrupts the quality of the secretome needs a little more discussion. Is it a direct change in the the quality control pathway, meaning the pathway being the substrate of these enzymes or the overall accumulation of other

inappropriately modified clients?

Response to point 2 and point 3: These two points are closely related and are briefly addressed together.

We have significant evidence that the acetylation status of nascent (trafficking) glycoproteins in the ER regulates their interaction with specific cargo receptors and transport machinery, at the intersection between conventional (CPS) and unconventional (UPS) protein secretion pathways. In other words, N ϵ -lysine acetylation of the nascent polypeptide appears to act as a signal that together with (but also independently of) the UPR directs the protein toward the CPS or UPS based on the folding status. This exciting project is currently ongoing and will be described in a follow-up paper.

The involvement of UPR, ERAD(I) and ERAD(II)/autophagy pathways has been described before in relevant mouse models (selected references are: PMID: PMC3436137; PMID: PMC4019794; PMID: PMC4805081; PMID: PMC4922439; PMID: PMC8041774).

Reviewer #3

The study implicates ATase 1 and ATase 2 in progeria phenotype. An excellent study done immaculately, with proper stats invoked. The study is slated for even better journals with clinical extrapolation.

Response: We are very grateful to the Reviewer for this positive evaluation of our study.

August 20, 2025

RE: Life Science Alliance Manuscript #LSA-2025-03378-TR

Prof. Luigi Puglielli
University of Wisconsin-Madison
Medicine
Waisman Center
1500 Highland Avenue
Madison, Wisconsin 53705

Dear Dr. Puglielli,

Thank you for submitting your revised manuscript entitled "Overexpression of ATase1 and ATase2 disrupts the secretome and causes a progeria phenotype". As you will see, Reviewer 2 is satisfied overall and has no further requests. Please consider clarifying the discussion in light of their comment on ATase1/2 impacts on cytosolic proteins. We invite you to include the figure provided for this reviewer validating mass spectrometry results as a supplementary figure to support these observations; however this is left to your discretion. We would be happy to publish your paper in Life Science Alliance pending final revisions necessary to meet our formatting guidelines.

- The titles in both the system and the manuscript file must be consistent with each other.
- Please use the [10 author names, et al.] format in your references (i.e., limit the author names to the first 10).
- Please add scale bars to the images in panels A and I of Figure 4.
- Please add molecular weight markers to the blots in Figure 8D.
- We encourage you to revise the figure legends for Figure S3 such that the figure panels are introduced in alphabetical order.
- Please add callouts for Figure 4D and G to your main manuscript text.
- Please remove Figures S7, S8 and S9 as well as the legends for these, and upload these as Supporting Data.
- We suggest you upload Figure 10 as a Graphical Abstract instead of as a Figure file.
- Please add ORCID ID for secondary corresponding author--they should have received instructions on how to do so.
- Please add the X and Bluesky handles of your host institute/organization, as well as your own and/or one of the authors, in our system.

A. FINAL FILES:

B. MANUSCRIPT ORGANIZATION AND FORMATTING:

Sincerely,

Reviewer #2 (Comments to the Authors (Required)):

In the revision, the authors have provided additional results to some of my concerns. The monitoring of COPII components is neglected. If the authors reason that the ER luminal enzymes studied here should not pose a direct impact on cytosolic factors, then they should simply mention this point in the response.

August 28, 2025

RE: Life Science Alliance Manuscript #LSA-2025-03378-TRR

Prof. Luigi Puglielli
University of Wisconsin-Madison
Medicine
Waisman Center
1500 Highland Avenue
Madison, Wisconsin 53705

Dear Dr. Puglielli,

Thank you for submitting your Research Article entitled "Overexpression of ATase1 and ATase2 disrupts the secretome and causes a progeria phenotype". Thank you for noting in your letter the ongoing work focused on protein acetylation in the ER and impacts in canonical and noncanonical secretion pathways. We understand your preference to retain this discussion for a future manuscript. We also appreciate your note on the western blot prepared for Reviewer 2. It is a pleasure to let you know that your manuscript is now accepted for publication in Life Science Alliance. Congratulations on this interesting work.

DISTRIBUTION OF MATERIALS:

Again, congratulations on a very nice paper. I hope you found the review process to be constructive and are pleased with how the manuscript was handled editorially. We look forward to future exciting submissions from your lab.

Sincerely,
